# Prevalence of traditional uvulectomy and its associated factors among parents who had less than 6 months of infants in Gondar City, north-west Ethiopia: Mixed study design

Lakachew Yismaw Bazezew[1]*, Asrat Hailu Dagne[2], Destaye Guadie Kassie[3], Abebaw Alemayehu Desta[4], Mengistu Berhan Gobeza[3]

1 Department of Pediatric and Neonatal Nursing, Debre Tabor University, Debre Tabor, Amhara Region, Ethiopia, 2 Department of Midwifery, Debre Tabor University, Debre Tabor, Amhara Region, Ethiopia, 3 Department of Pediatrics and Child Health, School of Nursing, University of Gondar, Gondar, Amhara Region, Ethiopia, 4 Department of Surgical Nursing, School of Nursing, University of Gondar, Gondar, Amhara Region, Ethiopia

* lakachewyismaw@gmail.com, Lakachewyismaw@dtu.edu.et

**Data Availability Statement:** Regarding data availability, there was an agreement between the Gondar City of Admiration Office, the Gondar City

## Abstract

### Background

Traditional uvulectomy is widely practiced in Africa especially in sub-Saharan countries including Ethiopia. Limited Studies conducted in different times and areas of the world have shown that the prevalence of uvulectomy and its associated factors are varied from country to country. There is limited evidence to determine the prevalence and reasons of malpractice in Ethiopia. Therefore, this study aimed to assess the prevalence of traditional uvulectomy, and its associated factors among parents who had children aged less than 6 months.

### Methods

A community-based cross-sectional mixed study design was conducted among 630 participants selected by a systematic sampling technique. Data were collected using a pretested interviewer-administered questionnaire. The collected data were entered into Epi-data, and analyzed by using STATA version 14software. Descriptive statistics were computed and presented using tables, figures and texts. Factors associated with the prevalence of uvulectomy were selected for multiple logistic regressions at the probability value (p-value) of less than 0.2 in the $X^2$- analysis. Statistically significant associated factors were identified at the probability value (p-value) less than 0.05 and adjusted OR (AOR) with a 95% CI. A total of 10 individual depth interviewees and 5 key informants were included using purposive sampling techniques. For the qualitative study, and thematic content analysis was employed to analyze the data, which were transcribed, translated, coded, screened, thematized, analyzed, synthesized, and finally supplemented the quantitative finding.

Health Office, the community leader and the author not to publish the raw data retrieved from the respondents' information. However, the datasets collected and analyzed for the current study can be obtained from the chief manager of the Gondar City health Office on behave of all selected Gondar City of Adumbrative kebeles for reasonable request (Name: Mr. Ayehu Gashe, Email:ayugashe@gmail. com., Phone: 251918788130).

**Funding:** The author(s) received no specific funding for this work.

**Competing interests:** The authors have declared that no competing interests exist.

## Results

The prevalence of traditional uvulectomy was 84.60% (CI: 81.77%n—87.42%). Being rural residence (adjusted OR = 2.08, CI = 1.06–4.12), mothers aged 30 to 34 years (adjusted OR = 2.9, CI = 1.28–6.60), having no antenatal care visit (adjusted OR = 3.91, CI = 1.33–11.47), having no bad obstetric history (adjusted OR = 2.11, CI = 1.12–3.98), having no postnatal care visit (adjusted OR = 3.88, CI = 1.36–11.12) and mothers' poor attitude (adjusted OR = 3.32, CI = 2.01–5.47) were statistically significant associated factors of traditional uvulectomy. Seeking traditional uvulectomy, lack of information and third-party pressure were the main themes of the qualitative study that support the findings of the quantitative study.

## Conclusion

The prevalence of traditional uvulectomy was high. Being rural residents, mothers aged 30 to 34 years, having no antenatal care visit, having no complications of obstetric history, having no postnatal care visit and mothers with poor attitude were statistically significant associated factors of traditional uvulectomy. Lack of information about uvulectomy and third-party pressure was the reason for seeking traditional uvulectomy.

## Background

Traditional uvulectomy is the total or partial removal of the uvula by the traditional healer and it is cutting of the uvula and sometimes the nearby structures like, tonsils [1–4]. A small soft tissue that hangs down from the back of the mouth above the throat of the two tonsils is the uvula. During the functional movement of the orpharyngeal isthmus, uvula is fundamental to the anatomical abort of the soft palate and It is a conical projection in the middle that extends from the soft palate's free edge [5, 6]. During the eleventh week, the two sections of the uvula fuse together in the midline to produce the soft palate [6]. Uvula influences the tone of voice, moistens the throat or pharyngeal mucosa, secrets large amount of thin serous saliva which lubricates the Oro-pharyngeal mucosa in conjunction with the soft palate, closes the nose-pharynx, and prevents aspiration/regurgitation of food or water through the nose which helps to prevent choking during swallowing, it boosts immunological function, and it is used in producing sounds necessary for communication [4, 7, 8].

Traditional instruments like sharp blades and threads are used for the cutting of the uvula [9–11]. The tools used to cut the uvula are not sterile and the procedure is conducted carelessly traditional ways, which exposes to a variety of health problems that could lead to local or systemic complications like speech problems, injury to the tongue, broken teeth, deformity of teeth, excessive bleeding lead to anemia and respiratory distress, long term dental problems, sepsis, tetanus, HIV/AIDSs, swallowing difficulty, malnutrition, otitis media, septicemia, and hepatitis [4, 12].

The prevalence of traditional uvulectomy varies in developing countries and it varies from 2.6% to 90% in Nigeria, Niger, and Ethiopia [11, 13–22]. The prevalence of infant mortality and morbidity are caused by different factors including immediate and chronic complications of traditional uvulectomy [23–25].

Different studies indicate that parents' and traditional practitioner's attitude, family income, being rural residents, having no antenatal care, antenatal and postnatal couple counseling about traditional uvulectomy, home delivery, being housewife, maternal

educational status, occupation, being rural residence, social outlook, parental perception towards uvulectomy, lack of effective modern treatment and third party pressure are among the factors that contribute to traditional uvulectomy practice [1, 13, 26, 27].

Despite efforts to reduce the problem over the years, gaps still exist. Findings from studies in different regions of Ethiopia showed that the prevalence is still high and many of the study participants reported that the victims are vulnerable to various health problems. Traditional uvulectomy can cause serious complications and the reason is cultural, which can lead to infant death, especially in sub-Saharan countries including Ethiopia [20, 27].

The recommendations to explore the factors related to traditional uvulectomy, and evidence related to traditional uvulectomy are limited. As a result, identifying the prevalence of traditional uvulectomy in infancy and its associated factors has public health importance. This study will help as an additional source of evidence after exploring local factors and the prevalence of the problem. It will contribute to improving the health of infants, changing parents' health behavior, and positive impacts on health institutions.

The traditional uvulectomy practice is cultural and varies from local to local or place to place and predictor variables like previous bad obstetric history were not considered so far. Moreover, the findings of this study will help program managers, stakeholders and health service providers to design appropriate interventions to prevent traditional uvulectomy. It will also be helpful to develop interventions aimed at reducing infant and child morbidity and mortality through increasing community awareness of uvulectomy as harmful traditional practice. Therefore, this study aimed to assess the prevalence of traditional uvulectomy and its associated factors among parents of children aged less than 6 months in Gondar City, North West Ethiopia, 2022.

## Methods

### Study design, period and setting

A community-based concurrent mixed study design was conducted from October 4 to November 4, 2022. The study was conducted in Gondar City, which is located 182 kilometers away from the regional city Bahir Dar, and 748 kilometers from the capital city Addis Ababa. Gondar is one of the ancient and densely populated Cities in Ethiopia and is located at an altitude of 12.4 North and 27.21 East. According to the recent administration, the city has 6 administration areas which consist of 36 kebeles. The city has 25 urban and 11 rural administrative kebeles. According to the 2023 Gondar city administrative estimation plan of vital registration, the total population was 475,172; among these 251, 466 and 223, 706 were females and males, respectively. The estimated number of mothers who had infants less than 6 months of age was 3136. The city has eight health centers, 50 private clinics, one private hospital, one public comprehensive specialized referral hospital, and one primary public hospital.

### Participants

Parents of infants less than 6 months who were living at least six months in Gondar City were considered for the quantitative study. Participants who were ill and unable to respond during the study period were excluded from the study. All mothers who had infants aged less than 6 months in Gondar City were the study population. The sample size was calculated using the single population proportion formula. The required sample size for this study was determined using the following assumptions; desired precision (d) = 5%, Confidence level = 95% ($Z\alpha/2 = 1.96$ value) and 52.5% of the prevalence of traditional uvulectomy in South Gondar Zone [13]. Hence, the calculated sample size by considering a 10% non-response rate was 634. Nine kebeles were selected from 36 kebeles of Gondar City through the lottery method, and based

on the total number of caregivers having children less than 6 months in each kebeles. The study sample proportional allocation was made. Finally, the participants were selected using systematic random sampling techniques after leveling of each household which had mothers who had children aged less than 6 months in each selected kebeles. Ten in-depth interviews from mothers who had less than 6 months old children and five key informants were included for the qualitative part of the study. Three health extension workers and 2 traditional were participated as key informants. The name of mothers, health extension workers and traditional healers was coded as I-01 to I-10, K-01 to K-03 and K-04 to K-05 respectively.

## Data collection tool and procedures

Socio-demographic and obstetric-related data collection tools comprised structured questionnaires that were prepared after a thorough literature review for the quantitative study. The local situation of the study area and the purpose of the study were considered for preparing the pretest. Questionnaires were developed first in English and then translated into Amharic which is the colloquial language of the respondents by language experts for ease of understanding of the respondents.

Senior nurses were engaged in data collection and they had previous experience in data collection. Training had been given to 4 data collectors and 2 supervisors for 3 days duration. It was focused on the questionnaire content to ensure consistency of data, obtaining consent, maintaining neutrality, privacy issues, personal relations and ethics in research. The pretest was carried out before the actual data collection period among 32 (5%) parents who had infants aged less than 6 months in Maksegnit Town. Based on the result of the pretest, corrections of confusion were made before the actual data collection time and data were collected via face-to-interviews. The filled questionnaires were checked daily for completeness and consistency of the responses to eliminate possible errors.

For the qualitative part of the study, the in-depth interview guide was prepared first in English then translated to Amharic and retranslated back to English for consistency. One data collector (research assistant) who had the educational status of master with previous qualitative data collection experience was selected. Both of research assistant and supervisor were trained to be familiar with the objective and the methodology of the research. The supervisor was considered as a research assistant for each session in addition to the assistant's roles. The supervisor participated as the overall activity controller when the principal investigator interviewed the participants. 15 respondents were coded and, categorized as key informants and individual depth informants, the interviews were audio-recorded and the interview duration was between 45 minutes and 60 minutes. The participants' emotions and non-verbal communication were taken as field notes. Saturation was determined when there were multiple overlapping responses across participants.

## Operational definition

**Traditional uvulectomy.**   Removal of the uvula by the traditional healer in the traditional way, this was measured from the report of parents and was coded 1 if parents reported that their infant had been subjected to traditional uvulectomy; otherwise, it was coded 0 [13, 14].

**Awareness.**   Awareness related issues of the respondent were assessed by five yes/no awareness-related questions which were measured separately from the report of caregivers and coded 1 if the caregivers reported yes; otherwise, it was coded 0 [1, 28].

**Attitude (Good, Poor).**   Those with greater than the mean score of 5 attitude-related questions were coded as having a good attitude and the rest as having a poor attitude [1, 13, 28].

## Data processing and analysis

The investigators and supervisors checked the data manually for completeness. Data were entered into EpiData version 3.1 and then exported to STATA version 14 for statistical analysis. Categorical variables were presented as frequency tables. Continuous variables were presented as descriptive measures, expressed as mean and standard deviation. A chi-square test was done to identify factors associated with the dependent variable. Independent variables with a probability value (P-value) of less than 0.2 in the Chi-square analysis were entered into the multivariable logistic regression model to identify the independent predictors of the traditional uvulectomy practice. Crude and adjusted odds ratios were used to identify the strength of the association at a 95% confidence interval. A P-value of less than 0.05 was used to decide the significance of the association.

Codes were created and described first for qualitative data. Categories were developed after defining the concepts. The categories were condensed by" collapsing the similar or dissimilar into broader higher-order categories". Finally, the codes were ordered into different themes, and the main theme, themes, and subthemes.

## Ethics approval and consent to participate

Ethical approval was obtained from the Ethical Committee of the University of Gondar, College of Medicine and Health Sciences with ethical code Ref.No: 037/2015, October 03/2022. A formal letter of cooperation was written for Gondar City health offices and kebeles, and permission to conduct the study was obtained from the offices and the Kebeles. Participants were informed that they had the right to withdraw from the study at any time. Moreover, we informed the purpose, procedures, advantages and disadvantages. Finally, informed written consent was obtained from each study participant.

## Result

### Socio-demographic characteristics

During the study period, a total of 630 mothers or caretakers who had infants aged less than 6 months were included in the study with a response rate of 99.36% and the non-response was due to incomplete information for some questionnaires. The mean age of mothers or caregivers and children was 29.68 years (SD ± 4.78) and 2.16 months (SD± 1.07), respectively. The majority of the respondents, 465 (73.81%), and 555 (88.10%) were from urban and Orthodox religious followers, respectively. About 92% of the participants were married. From the qualitative study of 15 respondents, 5 key informants, and 10 in-depth informants participated. Of all qualitative study participants, 9 of them were orthodox religious followers, and the rest of them were Protestants and Muslims. The remaining frequencies and percentages of socio-demographic-related characteristics of the quantitative study are presented on the next page (see Table 1).

### Obstetric related characteristics

Of the total participants, 83.97% (529) had one or more prenatal care visits. However, about 71.08% (376) of them were not counseled about the harmfulness of traditional infant uvulectomy. Regarding mothers' place of delivery, more than 99% of participants gave birth at a health facility. Of all the participants, 84.6% (533) of mothers had at least one postnatal care visit, but only 27.39% of them were counseled about the harmfulness of traditional uvulectomy. Moreover, the other obstetric, information and attitude-related factors were reported in the table (see Table 2).

**Table 1. Socio-demographic characteristics of parents of infants aged less than 6 months in Gondar City, Amhara Region, Ethiopia, 2022.**

| Variables | Frequency (n = 630) | Percent (%) |
|---|---|---|
| **0–1 month** | 423 | 67.1 |
| **Greater than one month** | 207 | 32.9 |
| Maternal or caretaker age | | |
| **18–24 year** | 80 | 12.7 |
| **25–29 year** | 247 | 39.21 |
| **30–34 year** | 188 | 29.84 |
| **Higher than or equal to 35 year** | 115 | 18.25 |
| Mothers' educational status | | |
| **Unable to read and write 12 grade and Diploma** | 43 | 6.83 |
| **Able to read and write** | 22 | 3.46 |
| **1st-12th grade** | 359 | 56.98 |
| **Diploma, degree and master holder** | 206 | 32.7 |
| Occupational status | | |
| **Farmer** | 52 | 8.25 |
| **Housewife** | 350 | 55.56 |
| **Civil servant** | 128 | 20.32 |
| **Daily laborer** | 33 | 5.24 |
| **Merchant** | 67 | 10.63 |
| Marital status | | |
| **Single** | 18 | 2.85 |
| **Married** | 584 | 92.7 |
| **Divorced** | 18 | 2.86 |
| **Widowed** | 10 | 1.59 |

## Practice of traditional uvulectomy

The prevalence of uvulectomy in the current study was 533(84.60%) (95%CI (81.77%-87.42%). The mean (±SD) age of the child when the uvula was cut is 4.90 (±2.4) days. Among those infants subjected to traditional uvula cutting, 85(15.94%) faced complications following the procedure and the complications were bleeding 42 (7.87%), wound (lesion) 36 (6.75%) and infectious disease 7 (1.3%). 429(80.49%) of subjected participants pay greater than 200 ETB for practitioners (See Table 3)

## Factors associated with traditional uvulectomy

According to the participants' responses, 84.6% (95% CI: 81.77% -87.42%) of infants were victims of traditional uvulectomy. Place of residence, age of mothers, Religion, educational status, antenatal care visit, PNC visits, bad obstetric history, heard harmfulness of uvulectomy, and mothers' attitude towards uvulectomy were the predictor variables found to be associated with traditional uvulectomy. These predictor variables were included in the multivariable model to identify the independent predictors of traditional uvulectomy.

The analysis revealed that participants being rural residents (adjusted OR = 2.08, CI = 1.06–4.12), mothers aged 30 to 34 years (adjusted OR = 2.9, CI = 1.28–6.60), having no antenatal care visit (adjusted OR = 3.91, CI = 1.33–11.47), having no bad obstetric history (adjusted OR = 2.11, CI = 1.12–3.98), having no postnatal care visit (adjusted OR = 3.88, CI = 1.36–11.12) and mothers' poor attitude (adjusted OR = 3.32, CI = 2.01–5.47) were statistically significant associated factors of traditional uvulectomy (See Table 4).

**Table 2. Obstetric, information and attitude-related characteristics of parents of infants aged less than 6 months in Gondar City, Amhara Region, Ethiopia, 2022.**

| Variables | | Frequency (n = 630) | Percent (%) |
|---|---|---|---|
| **Parity of mother** | | | |
| Primiparous | | 209 | 33.17 |
| Multiparous | 89.8 73 17.3 | 421 | 66.83 |
| **Had ANC follow-up** | | | |
| Yes | | 529 | 83.97 |
| No | | 101 | 16.03 |
| **Had PNC follow-up** | | | |
| Yes | | 533 | 84.6 |
| No | | 97 | 15.4 |
| **Had bad obstetric history** | | | |
| Yes | | 68 | 10.8 |
| No | | 562 | 89.2 |
| **Breastfeeding** | | | |
| Yes | | 374 | 59.4 |
| No | | 256 | 40.6 |
| **The attitude of participants towards uvulectomy** | | | |
| Good | | 211 | 33.5 |
| Poor | | 419 | 66.5 |
| **Hospitable health professional** | | | |
| Yes | | 85 | 13.5 |
| No | | 545 | 86.5 |
| **Heard harmfulness of uvulectomy** | | | |
| Yes | | 535 | 84.9 |
| No | | 95 | 15.1 |

ANC, Antenatal Care; PNC, Postnatal Care

The analysis of data from interviews generated three main themes. The three themes were seeking traditional uvulectomy, lack of information and third-party pressure. These main themes of the qualitative study support the findings of the quantitative study (see Table 5)

## Seeking traditional uvulectomy

This main theme, which defined seeking traditional uvulectomy, is a means to seek treatment for infants when they are sick. This theme supports the quantitative part of the study findings that indicate participants' poor attitude towards uvulectomy exposed the participant to seeking traditional uvulectomy, thinking low cost of uvulectomy and set aside the harmfulness of traditional uvulectomy. Therefore, they seek traditional uvulectomy. This is described by one of the interviewees as follows:

> *"We have information and know about traditional uvulectomy, even though traditional uvulectomy is better than the modern one. not only that, the cost for cutting ranges from 250 to 350 Ethiopian birr finally I recommend the government help the traditional healer by giving spatula, gloves, and new cutting materials to reduce the spread of infections (I-04)."*

**Table 3. Traditional uvulectomy practice of parents of infants aged less than 6 months in Gondar City, Amhara Region, Ethiopia, 2022.**

| Variables | Frequency (N = 630) | Percent (%) |
|---|---|---|
| **Traditional uvulectomy** | | |
| Yes | 533 | 84.60 |
| No | 97 | 15.40 |
| **Age of infant during uvulectomy childrenUUUUUUUpractice** | | |
| Within 7 days | 439 | 82.36 |
| Between 7 days and 6 months | 94 | 17.64 |
| **Cost of uvulectomy** | | |
| <200 ETB | 104 | 19.51 |
| ≥200ETB | 429 | 80.49 |
| **Complication** | | |
| Yes | 85 | 15.94 |
| No | 448 | 84.05 |
| **Types of complication** | | |
| Injury | 36 | 6.75 |
| Bleeding | 42 | 7.87 |
| Infection | 7 | 1.31 |

ETB = Ethiopian Birr

**Table 4. Bivariate and multivariable analysis of associated factors of traditional uvulectomy among parents of infants aged less than 6 months in Gondar City, Amhara Region, Ethiopia, 2022.**

| Variables | Uvulectomy | | Crude OR (95%CI) | Adjusted OR (95%CI) |
|---|---|---|---|---|
| | **Yes** | **No** | | |
| **Place of residence** | | | | |
| Urban | 382 | 83 | 1 | |
| Rural | 151 | 14 | ***2.34 (1.29–2.25) | *2.08 (1.08–4.12) |
| **Age of mothers** | | | | |
| 20–24 year | 62 | 18 | 1 | |
| 25–29 year | 203 | 43 | **1.37(0.74–2.56) | 1.66(0.79–3.49) |
| 30–34 year | 164 | 20 | ***2.43(1.21–4.91) | **2.9(1.28–6.6) |
| Higher than or equal to 35 year | 65 | 11 | **1.79(0.85–3.78) | 2.12(0.90–4.97) |
| **Had ANC follow-up** | | | | |
| Yes | 436 | 93 | 1 | |
| No | 97 | 4 | ***5.17(1.85–14.41) | **3.91(1.33–11.4) |
| **Had bad obstetric history** | | | | |
| Yes | 46 | 22 | 1 | |
| No | 487 | 75 | **3.11(1.76–5.45) | *2.11(1.12–3.98) |
| **Had PNC follow-up** | | | | |
| Yes | 441 | 92 | 1 | |
| No | 92 | 5 | ***3.83(1.52–9.71) | **3.88(1.36–11.1) |
| **Mothers' attitude toward uvulectomy** | | | | |
| Good | 150 | 61 | 1 | |
| Poor | 383 | 36 | ***4.33(2.75–6.81) | ***3.3(2.01–5.47) |

*, P Value≤0.05; **, P Value≤0.01;**, P Value≤0.001

ANC, Antenatal Care; PNC, Postnatal Care

**Table 5. Summary of themes for traditional uvulectomy practice in Gondar city, Amhara region, Ethiopia, 2022.**

| Main theme | Seeking traditional uvulectomy | Lack of information | Third-party pressure |
|---|---|---|---|
| Themes | • Signs, symptoms, and complications of uvulitis<br>• Access to health services | • Information on the uvula, and uvulectomy<br>• Perception of uvula and uvulectomy | • Family influence<br>• Community influence |
| Subthemes | • Fear Signs, symptoms<br>• Fear of Complications, and sever illness of the uvulitis<br>• Poor coverage of healthcare services<br>• Poor behavioral change | • Lack of awareness of uvula<br>• Lack of awareness of uvulectomy<br>• Poor Attitude about uvula<br>• Poor attitude about uvulectomy | • Fathers influence<br>• Grandparents influence<br>• Peer influence<br>• Community leader influence<br>• Religious leaders influence |

## Lack of information

One of the health extension workers claimed that lack of information is one of the issues of traditional uvulectomy. She described that educating and training the community to boost knowledge on harmful traditional uvulectomy is crucial. This main theme supports the findings of a quantitative study that discloses having no ANC and PNC are the factors increasing traditional uvulectomy due to lack of getting health education on traditional uvulectomy during ANC and PNC visits. This is explained by one of the key informants:

"*In my opinion, the first prevention modalities are by educating and training the community. Second, if the event is proven as bad, legal action should be taken by the responsible body against individuals who are not changed through education. In another way, most people go to health institutes to treat their children, but infants may not be cured. Mothers want to be cut it due to the reason, which if not mothers desire to continue the treatment (if the problems of their child are not cured by medical care at a health institution, mothers going to get care from a traditional healer). After three to four modern treatment options mothers were also going to cut it traditionally. If not cured by the modern one, they choose the traditional one (k-01)."*

## Third-party pressure

Third-party pressure is defined as the motivation of an individual or community to participate in ceremonies or events for the benefit of other parties. One of the interviewees described that the third-party pressure leads to traditional uvulectomy as follows:

"*Today's modern treatment option cannot treat and prevent the problem after and before it occurs*", "*As I usually heard from my neighbors*" when uvula elongated and swelled will lead to child death. My family and neighbors put pressure on me and at that time, we seek to be cut the uvula (I-09)."

## Discussion

The prevalence of traditional uvulectomy was determined and statistically significant associated factors of traditional uvulectomy were identified.

The prevalence of traditional uvulectomy in this study was higher than the finding of a study conducted in Axum, Ethiopia [14], Debre Birhan, Ethiopia, 23.7% [26], Debre Tabor comprehensive specialized hospital,15.88%(1), in Khartoum hospitals, 17.9% [29] and Nigeria [11, 18]. This difference may be due to the difference in the study periods which is greater than the expected gaps that are longer than 5 years between the conducted studies, and the study participants of the current study whose infants aged less than 6 months rather than most other

study groups whose under 5 years children, Which were less commonly exposed group for traditional uvulectomy.

The prevalence of traditional uvulectomy in this study was in line with a study conducted in Plateau State Nigeria [30], and in East Shewa Zone at Fentale Woreda, Ethiopia [31]. The similarity may be due to the study setting, and unimproved awareness creation, inadequate access and quality of healthcare service and it may also be due to the related situations of communities' attitudes and healthcare service. The prevalence of traditional uvulectomy for this study is lower than the study conducted in Jigawa state Nigeria. The possible reason for the difference may be due to the difference in study periods which is longer than 5 years between the two studies. This might also be because of health-seeking behavior, lack of health facilities, quality health care and better development of infrastructure.

The odds of practicing traditional uvulectomy among participants of rural residence were 2.08 times more likely than participants of urban residence. A similar result was found in a study conducted in Ethiopia [16, 27]. The possible reason for this may be the study setting which causes lower level of awareness and poor attitudes because of poor access to health care services and lack of access to social media towards traditional uvulectomy, on the contrary easily accessible to traditional healers.

This study revealed that the odds of traditional uvulectomy among participants aged 30 to 34 years were 2.9 times higher than participants aged 20–24 years. This may be because most participants in this age group are more prone to adopt the experiences of traditional malpractices from the community and their poor attitude considering the advantages outweigh its bad effects, also outweigh previous good results of malpractice than bad effects of it.

Mothers who had no ANC visit were almost 3.91 times more likely to have infants suffering from traditional uvulectomy than those who had ANC visit. This finding is similar to the study conducted in Nigeria and Ethiopia [1,15, 27]. This study result is also supported by the qualitative result of this study since the majority of the participants practiced traditional uvulectomy due to lack of information on the effects of malpractice. This may also be due to not being counseled towards traditional uvulectomy, visiting health institutions, and poor attitude towards health facility services.

The odds of traditional uvulectomy among mothers of infants who had no bad obstetric history were 2 times higher than mothers of infants who had bad obstetric history. This could be because of no repeated visits to health institutions for the bad obstetric history that may not increase the awareness and attitude during ANC follow-up to prevent uvulectomy. Since, they were less likely to visit health institutions; it is true that there is less opportunity to be counseled towards traditional uvulectomy.

The odds of traditional uvulectomy practiced among mothers who had no PNC visit were 3.88 times higher than mothers who had a PNC visit. The finding of this study is consistent with the study conducted in Ethiopia [1]. This study result is also supported by the qualitative part of this study that revealed the majority of the participants practiced traditional uvulectomy due to their unscientific perceptions and the lack of information about the malpractice. This is true that if there are no repeated visits to health institutions, mothers may not increase their awareness and attitude level to prevent the experience of uvulectomy. In addition to this, it could be because of less opportunity to be counseled towards traditional uvulectomy.

Poor attitude of mothers of infants aged less than 6 months was a statistically significant associated factor of traditional uvulectomy. This study coincides with a study carried out in Ethiopia [16]. This could be due to the mothers' perceptions towards traditional uvulectomy practice which pushes them to accept an effective alternative treatment modality as a health care service. Thus the attitude of parents is crucial to decide which treatment option is best to treat their infant's illness.

Seeking traditional uvulectomy, lack of information and third-party pressure were the main themes as reasons to practice traditional uvulectomy in this qualitative study. This qualitative study result supported the quantitative findings of this study. The lack of information about the treatment of uvulitis described in this qualitative study finding is supported by the findings of a qualitative study conducted in Dadar, Oromia region [32].

Third-party pressure is described as the reason for traditional uvulectomy practice and this finding is also supported by a qualitative study conducted in Dadar, Oromia [32]. The similarity may be due to unimproved awareness creation, inadequate access and quality of health care service and it may also be due to the related situations of communities' attitudes and healthcare service of traditional uvulectomy.

## Limitation

There could be the possibility of recall bias because mothers were asked to report some events of associated factors within months sometimes ago.

## Conclusion

The prevalence of traditional uvulectomy was high. Being rural mothers, mothers aged 30 to 34 years, having no antenatal care visit, having no complications of obstetric history, having no postnatal care visit and mothers with poor attitude were statistically significant associated factors of traditional uvulectomy. Lack of information about uvulectomy and third-party pressure was the reason for seeking traditional uvulectomy. The findings of this study will serve as a baseline for intervention of traditional uvulectomy practice. Therefore, it is better to increase the awareness of parents and the community, and change the attitude and health seeking behavior of parents through advice, counseling and education during antenatal and postnatal care services.

## Acknowledgments

We are thankful to the data collectors and all the study participants for their willingness to participate in the study. We would also like to thank the University of Gondar College of Medicine and Health Science School of Nursing for the facilitation of the program. We would also like to thank Debre Tabor University, College of Health Sciences, department of pediatrics and neonatal nursing for the support during data collection.

## Author Contributions

**Conceptualization:** Lakachew Yismaw Bazezew, Destaye Guadie Kassie, Mengistu Berhan Gobeza.

**Data curation:** Lakachew Yismaw Bazezew, Destaye Guadie Kassie, Abebaw Alemayehu Desta, Mengistu Berhan Gobeza.

**Formal analysis:** Lakachew Yismaw Bazezew, Destaye Guadie Kassie, Abebaw Alemayehu Desta.

**Funding acquisition:** Destaye Guadie Kassie.

**Investigation:** Lakachew Yismaw Bazezew.

**Methodology:** Lakachew Yismaw Bazezew, Asrat Hailu Dagne, Destaye Guadie Kassie, Abebaw Alemayehu Desta, Mengistu Berhan Gobeza.

**Software:** Lakachew Yismaw Bazezew.

**Supervision:** Lakachew Yismaw Bazezew, Asrat Hailu Dagne, Destaye Guadie Kassie, Abebaw Alemayehu Desta.

**Validation:** Lakachew Yismaw Bazezew, Asrat Hailu Dagne, Destaye Guadie Kassie, Abebaw Alemayehu Desta.

**Visualization:** Lakachew Yismaw Bazezew, Asrat Hailu Dagne, Destaye Guadie Kassie, Abebaw Alemayehu Desta.

**Writing – original draft:** Lakachew Yismaw Bazezew, Asrat Hailu Dagne, Destaye Guadie Kassie, Abebaw Alemayehu Desta, Mengistu Berhan Gobeza.

**Writing – review & editing:** Lakachew Yismaw Bazezew, Asrat Hailu Dagne, Destaye Guadie Kassie, Abebaw Alemayehu Desta, Mengistu Berhan Gobeza.

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
