## [Decision Letter · Decision Letter 0]

1 Mar 2024

PONE-D-23-22864Prevalence of traditional uvulectomy and its associated factors among parents who had less than 6 months of infants in Gondar city, north-west Ethiopia: Mixed study designPLOS ONE

Dear Dr. Bazezew,

Thank you for submitting your manuscript to PLOS ONE. After careful consideration, we feel that it has merit but does not fully meet PLOS ONE’s publication criteria as it currently stands. Therefore, we invite you to submit a revised version of the manuscript that addresses the points raised during the review process.

Please submit your revised manuscript by Apr 15 2024 11:59PM**.** If you will need more time than this to complete your revisions, please reply to this message or contact the journal office at plosone@plos.org. Please include the following items when submitting your revised manuscript:A rebuttal letter that responds to each point raised by the academic editor and reviewer(s). You should upload this letter as a separate file labeled 'Response to Reviewers'.A marked-up copy of your manuscript that highlights changes made to the original version. You should upload this as a separate file labeled 'Revised Manuscript with Track Changes'.An unmarked version of your revised paper without tracked changes. You should upload this as a separate file labeled 'Manuscript'.

We look forward to receiving your revised manuscript.

Kind regards,

Kahsu Gebrekidan

Academic Editor

PLOS ONE

Journal Requirements:

3. In this instance it seems there may be acceptable restrictions in place that prevent the public sharing of your minimal data. However, in line with our goal of ensuring long-term data availability to all interested researchers, PLOS’ Data Policy states that authors cannot be the sole named individuals responsible for ensuring data access (http://journals.plos.org/plosone/s/data-availability#loc-acceptable-data-sharing-methods).

Reviewers' comments:

Reviewer's Responses to Questions

**Comments to the Author**

1. Is the manuscript technically sound, and do the data support the conclusions?

Reviewer #1: Yes

Reviewer #2: Yes

2. Has the statistical analysis been performed appropriately and rigorously? 

Reviewer #1: Yes

Reviewer #2: Yes

3. Have the authors made all data underlying the findings in their manuscript fully available?

Reviewer #1: Yes

Reviewer #2: No

4. Is the manuscript presented in an intelligible fashion and written in standard English?

Reviewer #1: Yes

Reviewer #2: Yes

5. Review Comments to the Author

Reviewer #1: Title: The title could be shorter and stated as follows:

"Prevalence of traditional uvulectomy and its associated factors in Gondar city, north-west Ethiopia: Mixed study design"

And in the work method, parents with babies less than 6 months old should be included as entry criteria.

Abstract: Method: Explain about the data collection tool

Line 43: In the abstract it is not necessary to say "escriptive statistics were computed and presented using tables, figures and texts."

Background: Write the introduction a little more briefly

Methods:

Line 180-182: In the quantitative part of the study, how many questions did the data collection tool have? Except for demographic characteristics, about other components such as knowledge and attitude, if there is a question, state the items.

Line 183: Explain how to score the questions and the validity and reliability of the data collection tool

Line 196: For qualitative part of the study, the in-depth interview guide what parts did it include?

Line 201: Where was the location of the interviews?

Line 228: How many people coded the qualitative data and categorized them?

Result

Line 347: In a table, report the main and sub-themes from the qualitative study.

Discussion

It would have been better to have at least one paragraph discussing only the results of the qualitative study

Conclusion

In the conclusion, suggestions and solutions based on the results of the present study should be presented

The ethics approval code should be written

Reviewer #2: Title: Prevalence of traditional uvulectomy and its associated factors among parents who

had less than 6 months of infants in Gondar city, north-west Ethiopia: Mixed study

design

The title is in line with study objectives and method

Abstract: Well structured

• Minor typographical error on line 40.

Background: Detailed information on statement of problem, rational for the study and study objectives are clearly.

• Some grammatical errors highlighted for correction.

Methods: Fairly well described.

• Line 175 - Selection of participants by systematic sampling - This is not clear.

• Was there a sampling frame for the communities? This is a community-based study. It is unlikely to select respondents in a city without the use of multi-stage random sampling.

• Provide the total number of KIIs and how they were distributed among health extension workers and traditional healers.

• Was interview conducted among the nursing mothers?

• Line 198 - Does this mean only one research assistant was used for qualitative study. Qualitative study requires at least a pair of research assistants for each session.

Result: Fairly well written in details with relevant tables and figures.

• Line 332 - Which table provided this data? No table clearly showed the prevalence of traditional uvulectomy.

• Line 372 – The highlighted statement is not clear

Discussion: The study findings are well discussed and compared with other studies.

• Prevalence of uvulectomy not clearly shown on any of the tables.

• Check the spelling, grammatical and syntax errors highlighted for correction.

Conclusion: Clearly written but without any clear recommendation based on the findings in this study.

6. PLOS authors have the option to publish the peer review history of their article (what does this mean?). If published, this will include your full peer review and any attached files.

Reviewer #1: **Yes: **Masoumeh Alidosti

Reviewer #2: **Yes: **Prof. Tanimola Makanjuola Akande

---

## [Author Response · Author response to Decision Letter 0]

2 May 2024

PLOS/One

From Lakachew Yismaw Bazezew

E- mail: lakachewyismaw@gmail.com(1)

Subject: submitting response to reviewers’, and academic editor’s comments and questions 

Title: - Prevalence of traditional uvulectomy and its associated factors among parents who had less than 6 months of infants in Gondar city, north-west Ethiopia: Mixed study design

First of all, we would like to thank academic editor and reviewers for your indispensable comments and supports to forward our manuscript. We included all constructive academic editor’s and reviewers’ comments. We have revised all things made point by point in accordance with the comments and questions of academic editor and reviewers. Please notice that we included the responses to each reviewer’s and academic editor’s comments and questions in the following responses.

Response to academic editor’s Comments

� Editor’s comment: Please ensure that your manuscript meets PLOS ONE's style requirements, including those for file naming. The PLOS ONE style templates can be found at https://journals.plos.org/plosone/s/file?id=wjVg/PLOSOne_formatting_sample_main_body.pdf andhttps://journals.plos.org/plosone/s/file?id=ba62/PLOSOne_formatting_sample_title_authors_affiliations.pdf

� Authors’ response: We thank academic editor, and we ensured that our manuscript meets PLOS ONE's style requirements, including those for file naming. 

� Editor’s comment: We suggest you thoroughly copyedit your manuscript for language usage, spelling, and grammar. If you do not know anyone who can help you do this, you may wish to consider employing a professional scientific editing service. 

� Authors’ response: We thank academic editor, and we employed language and professional editor for scientific editing service to correct our manuscript for language usage, spelling, and grammar (see at the next table )

Name Of Editor Institution Phone Email Department 

Asrat Hailu Dagne1 and Dr. Yalew Aklog Aweke2

1. 1Department of Midwifery, Debre Tabor University, Debre Tabor, Amhara Region, Ethiopia

Email Address: 1221asrat@gmail.com Phone: +251912196204

2. Department of language, Bahir Dar University, Bahir Dar, Amhara Region, Ethiopia

Email Address: aklogyalew@yahoo.com Phone: +251921252375 

Editor’s comment: there may be acceptable restrictions in place that prevent the public sharing of your minimal data. However, in line with our goal of ensuring long-term data availability to all interested researchers, PLOS’ Data Policy states that authors cannot be the sole named individuals responsible for ensuring data access (http://journals.plos.org/plosone/s/data-availability#loc-acceptable-data-sharing-methods). Data requests to a non-author institutional point of contact, such as a data access or ethics committee, helps guarantee long term stability and availability of data. Providing interested researchers with a durable point of contact ensures data will be accessible even if an author changes email addresses, institutions, or becomes unavailable to answer requests. Before we proceed with your manuscript, please also provide non-author contact information (phone/email/hyperlink) for a data access committee, ethics committee, or other institutional body to which data requests may be sent. If no institutional body is available to respond to requests for your minimal data, please consider if there any institutional representatives who did not collaborate in the study, and are not listed as authors on the manuscript, who would be able to hold the data and respond to external requests for data access? If so, please provide their contact information (i.e., email address). Please also provide details on how you will ensure persistent or long-term data storage and availability.

Authors’ response: There was an agreement between Gondar city of admiration office, Gondar city health office, community leader and author not to share the raw data retrieved from the respondents’ information for third party.

However, you can contact Mr. Belachew Golla, Email:belachewgoll@gmail.com and discuss to share the datasets collected and analyzed for the current study

Editor’s comment: PLOS requires an ORCID iD for the corresponding author in Editorial Manager on papers submitted after December 6th, 2016. Please ensure that you have an ORCID iD and that it is validated in Editorial Manager. To do this, go to ‘Update my Information’ (in the upper left-hand corner of the main menu), and click on the Fetch/Validate link next to the ORCID field. This will take you to the ORCID site and allow you to create a new iD or authenticate a pre-existing iD in Editorial Manager. Please see the following video for instructions on linking an ORCID iD to your Editorial Manager account: https://www.youtube.com/watch?v=_xcclfuvtxQ

Authors’ response: we thank academic editor , and we ensured an ORCID ID, that it is validated in Editorial Manager to fulfill PLOS requirement for the corresponding author in Editorial Manager on papers submitted after December 6th, 2016 Editorial Manager. To do this, I have gone to ‘Update our Information’ (in the upper left-hand corner of the main menu), and have clicked on the Fetch/Validate link next to the ORCID field. This took us to the ORCID site and allowed us to create a new ID or authenticate a pre-existing ID in Editorial Manager. 

Responses to reviewers’ Comments and questions 

Responses to reviewer 1

Reviewer’s comments: The title could be shorter and stated as follows: "Prevalence of traditional uvulectomy and its associated factors in Gondar city, Northwest Ethiopia: Mixed study design”

Authors’ response: We thank reviewer’s ( #1) constructive comment , and Considered reviewer’s recommendation, we considered both of reviewer’s comments and the manuscripts objectives of the title, and we have used the previous title as it was and the second reviewer recommended that the title continue as it was (Prof. Tanimola Makanjuola Akande or Reviewer #2) : Title: Prevalence of traditional uvulectomy and its associated factors among parents who had less than 6 months of infants in Gondar city, north-west Ethiopia: Mixed study Design (see page 1, line 1-3)

Reviewer’s comments and questions: in the work method, parents with babies less than 6 months old should be included as entry criteria.

Authors’ response: we appreciate reviewer’s comment and we accepted the comment we already considered parents with babies less than 6 months old have been included as entry criteria in the work method.(see page 7,Line:162-163

Reviewer’s comment: Abstract: Method: Explain about the data collection tool

Authors’ response: we thank reviewer for your comment, and we agreed with reviewer’s comment, and we have explained the data collection tool at the method section of abstract.(see page 2,Line:40-41)

Reviewer’s comments: In the abstract it is not necessary to say "escriptive statistics were computed and presented using tables, figures and texts."

Authors’ response: We thank you very much, and appreciated reviewer’s for their valuable focus to correct, and forward our manuscript and we corrected capitalization, spelling error in the abstract the word “escriptive” to descriptive. (see page 2,Line:39 and 43)

Reviewer’s comments :Background: Write the introduction a little more briefly

Authors’ response: We thank you, agreed with reviewer comment, and we wrote the introduction a little more briefly.( see page 4,Line:79-83)

Reviewer’s suggestion and questions: In the quantitative part of the study, how many questions did the data collection tool have? Except for demographic characteristics, about other components such as knowledge and attitude, if there is a question, state the items.

Authors’ response: We accepted reviewer comment and addressed question as follows:, there are main questions which are infrastructure related question like: 1.no nearby health institution, 2.lack of better medical care, 3.presence of traditional practitioner, obstetric related questions like: 1.number of ANC visits, 2.Antenatal counseling of traditional uvuloctomy, 3.ever had bad obstetrics history, 4.Place of delivery, 5.Number of PNC visits, 6.Postnatal counseling about traditional uvulectomy, 7.Parity of mother, 8.ever had Breastfeeding ,9. Hospitable health professional. Awareness related questions such as: 1.Information about traditional uvuloctomy, 2.Source of information about uvulectomy, 3.Information about uvula, 4.source of information about uvula 5.information about benefit of uvulectomy, practice related question such as: 1.ever had Traditional uvulectomy, 2. Age of infant during uvulectomy practice, 3. Cost of uvulectomy, 4.Complication of uvulectomy, 5.Types of complication, and Attitude related questions, such as: 1. Do you think that Uvulectomy causes disease or illness? 2. Do you think that traditional Uvulectomy is harmful? Do you perform traditional uvulectomy in the future? 4. Do you encourage other people to cut their baby’s uvula? 5. Do you think that uvula cutting should be eradicated? 6. Reason to seek traditional uvulectomy.

Reviewer’s questions: Explain how to score the questions and the validity and reliability of the data collection tool?

Authors’ response: To address reviewers recommendation and questions the way of scoring the questions are as following: Awareness: awareness related issues of the respondent was assessed by five yes/no awareness related questions which was measured separately from the report of caregivers and coded 1 if the caregivers reported yes for the given awareness related information; otherwise, it was coded 0 for an unwearied information. Attitude (Good, Poor): The attitude of the participants toward Uvulectomy was assessed using 5 questions with yes, and no responses. At first, questions 3 and 4 of attitude questions were recoded (1 to 0 and 2 to 1) to bring similar responses/meanings/, Then those 5 questions were computed for making one category (attitude) and attitude was grouped into two (good and poor) using the mean score as a cutoff point. Those with less than or equal to the mean score of 5 yes /no attitude related questions were coded as having a good attitude and the rest as having a poor attitude. Validity and reliability of data collection tool insured by choose appropriate tools, Designed and tested instruments, Implemented and monitored instruments, analyzed and reported instruments. We used revalidated tools taken from different literatures. Selected most suitable research question, designed, and selected representative population to ensured validity and reliability of data collection tools. We considered the type, level, and scope of data that we need, as well as the feasibility, cost, and ethics of using different tools. We have used pretest through closed-ended questions that can be easily administered and analyzed. On the other hand when we want to explore the experiences and meanings of informants, we used an interview with open-ended questions that can elicit rich and detailed responses for qualitative parts of study. We Designed and tested our instruments carefully before using them in our actual research to ensure the validity and reliability of our data collection tools. We have used clear and simple language, avoided bias and ambiguity, provided instructions and we have tested our tools on a sample of our target population or experts in our field to check for any errors, misunderstandings, or difficulties that might affect the validity and reliability of our data. We have used techniques such as pretest and expert review to evaluate and improve our instruments. We Implemented and monitored our instruments in a consistent and standardized way in our actual research to ensure the validity and reliability of our data collection tools. such as selecting a representative and random sample, obtaining informed consent, providing clear and accurate instructions, maintaining rapport and confidentiality, and avoided interference or influence. We also monitored our instruments throughout the data collection process, such as recording any problems, deviations, or feedback that might affect the validity and reliability of our data. We have used methods such as quality control, quality assurance, to check and ensure the quality of our data collection tool. We Analyzed and reported our tool in a transparent and rigorous way in our research report to ensure the validity and reliability of our data collection instruments. We have used appropriate and valid statistical methods to analyze our data, such as descriptive statistics, and factor analysis. We have acknowledged any limitations, challenges that might affect the validity and reliability of our data collection tools, and suggested ways to overcome or address them in future research.

Reviewer’s questions: For qualitative part of the study, the in-depth interview guide what parts did it include? What types of interview is this? How is it different from KIIs?

Authors’ response: we thank and appreciated for support to forward our manuscript, to address reviewer question, it has two parts, the first part is engaging ,and the second is probing Semi-structured guiding questioners which was applied for key and individual in-depth informants accordingly.to clarify their deference ,individual depth interview focused on the gathering information from individuals own self experience, reasons and perceptions to their real practice of the study participants ,and KIIs is interview that focused on the experience, reason ,and perceptions of the practice on behalf of the study participants, and have the knowledge and practice on the phenomenon . 

Reviewer’s questions: Where was the location of the interviews?

Authors’ response: we appreciated reviewer and to address reviewer’s question, and the interview was conducted at participants home that was free from any disturbance), audio recording sound, and time frames were maintained appropriately. 

Reviewer’s comments and questions: How many people coded the qualitative data and categorized them?

Authors’ response: Ten individual depth interviews from mothers who had less than 6 months old children and five key informants were included for qualitative part of the study, Three health extension workers and two traditional healers were participated as key informants. Name of mothers, health extension workers and traditional healers was coded as I-01 to I-10, K-01 to K-03 and K-04 to K-05 respectively.(see page 8,Line:175 to 180)

Reviewer’s comments and questions: In a table, report the main and sub-themes from the qualitative study. 

Author’s response:

Thank you for your constructive comment, and we accepted the comment and reported the main and sub-themes in a table at main document.(see page 21,Line:434 to 436 )

Reviewer’s comments and questions: It would have been better to have at least one paragraph discussing only the results of the qualitative study. 

Authors’ response: We would like to thank reviewer for your constructive comments, and we had discussed only the result of qualitative study in a paragraph at discussion part. (see page 24-25,Line: 505 -515

Reviewer’s comments: In the conclusion, suggestions and solutions based on the results of the present study should be presented.

Authors’ response: We accepted the reviewer constructive comments and we presented suggestions and solutions based on the results of the present study at main documents.(see page25 ,line 525-528)

Reviewer’s comment: It would have been better to have at least one paragraph discussing only the results of the qualitative study

Authors’ response: We accepted and appreciated the reviewer comment and the reference is updated at discussion part of manuscript which was available in the main documents.( see page22 ,line 448 to 450 )

Reviewer’s comment: The ethics approval code should be written

Authors’ response: We accepted reviewer comments, and appreciated your support, we have written ethics approval code at main documents at ethics parts of it as Ref.N.o:037/2015) october03/2022.(see page10 ,Line:241-242)

Response to reviewer 2 

Reviewer’s comments: Title: Prevalence of traditional uvulectomy and its 

---

## [Decision Letter · Decision Letter 1]

27 May 2024

Prevalence of traditional uvulectomy and its associated factors among parents who had less than 6 months of infants in Gondar City, north-west Ethiopia: Mixed study design

PONE-D-23-22864R1

Dear Mr Lakachew,

We’re pleased to inform you that your manuscript has been judged scientifically suitable for publication and will be formally accepted for publication once it meets all outstanding technical requirements.

Kind regards,

Kahsu Gebrekidan

Academic Editor

PLOS ONE

Additional Editor Comments (optional):

Reviewers' comments:

Reviewer's Responses to Questions

**Comments to the Author**

1. If the authors have adequately addressed your comments raised in a previous round of review and you feel that this manuscript is now acceptable for publication, you may indicate that here to bypass the “Comments to the Author” section, enter your conflict of interest statement in the “Confidential to Editor” section, and submit your "Accept" recommendation.

Reviewer #1: All comments have been addressed

Reviewer #2: All comments have been addressed

2. Is the manuscript technically sound, and do the data support the conclusions?

Reviewer #1: Yes

Reviewer #2: (No Response)

3. Has the statistical analysis been performed appropriately and rigorously? 

Reviewer #1: Yes

Reviewer #2: (No Response)

4. Have the authors made all data underlying the findings in their manuscript fully available?

Reviewer #1: Yes

Reviewer #2: (No Response)

5. Is the manuscript presented in an intelligible fashion and written in standard English?

Reviewer #1: Yes

Reviewer #2: (No Response)

6. Review Comments to the Author

Reviewer #1: (No Response)

Reviewer #2: (No Response)

7. PLOS authors have the option to publish the peer review history of their article (what does this mean?). If published, this will include your full peer review and any attached files.

Reviewer #1: No

Reviewer #2: **Yes: **Prof. Tanimola Makanjuola AKANDE

---

## [Editor Report · Acceptance letter]

31 May 2024

PONE-D-23-22864R1 

PLOS ONE

Dear Dr. Bazezew, 

I'm pleased to inform you that your manuscript has been deemed suitable for publication in PLOS ONE. Congratulations! Your manuscript is now being handed over to our production team.

Kind regards, 

on behalf of

Dr. Kahsu Gebrekidan 

Academic Editor

PLOS ONE